# Differentiating Real-World Autobiographical Experiences without Recourse to Behaviour

**DOI:** 10.3390/brainsci11040521

**Published:** 2021-04-20

**Authors:** Jonathan Erez, Marie-Eve Gagnon, Adrian M. Owen

**Affiliations:** 1The Brain and Mind Institute, Western University, London, ON N6A 3K7, Canada; marie-eve.gagnon@uqtr.ca (M.-E.G.); aowen6@uwo.ca (A.M.O.); 2Department of Physiology and Pharmacology, Western University, London, ON N6A 5C1, Canada; 3Department of Psychology, Western University, London, ON N6A 5C2, Canada

**Keywords:** autobiographical memory, fMRI, wearable camera, fronto-parietal network, machine-learning

## Abstract

Investigating human consciousness based on brain activity alone is a key challenge in cognitive neuroscience. One of its central facets, the ability to form autobiographical memories, has been investigated through several fMRI studies that have revealed a pattern of activity across a network of frontal, parietal, and medial temporal lobe regions when participants view personal photographs, as opposed to when they view photographs from someone else’s life. Here, our goal was to attempt to decode when participants were re-experiencing an entire event, captured on video from a first-person perspective, relative to a very similar event experienced by someone else. Participants were asked to sit passively in a wheelchair while a researcher pushed them around a local mall. A small wearable camera was mounted on each participant, in order to capture autobiographical videos of the visit from a first-person perspective. One week later, participants were scanned while they passively viewed different categories of videos; some were autobiographical, while others were not. A machine-learning model was able to successfully classify the video categories above chance, both within and across participants, suggesting that there is a shared mechanism differentiating autobiographical experiences from non-autobiographical ones. Moreover, the classifier brain maps revealed that the fronto-parietal network, mid-temporal regions and extrastriate cortex were critical for differentiating between autobiographical and non-autobiographical memories. We argue that this novel paradigm captures the true nature of autobiographical memories, and is well suited to patients (e.g., with brain injuries) who may be unable to respond reliably to traditional experimental stimuli.

## 1. Introduction

One of the key facets of human consciousness is autobiographical memory. This ability, to form and later re-experience personal events is, to a large extent, what gives us a sense of our personal identity. However, because autobiographical memory can only be tested through self-report, it is vulnerable to distortion, intentionally or otherwise, by those doing the reporting. Moreover, in some patient groups, the ability to report may be compromised, precluding the examination of autobiographical memories altogether. An extant challenge for cognitive neuroscience, therefore, is to decode the contents of memories from brain activity alone, without recourse to any behavioural response. In recent years, several studies have shown that similar conscious experiences elicit a common pattern of neural activity across individuals [1,2], suggesting a solution to this challenge. However, autobiographical memories are inherently personal, and cannot easily be compared from one individual to another. In this study, we recorded real-life autobiographical events in a naturalistic setting and investigated whether the fMRI responses elicited while viewing these personal experiences can be differentiated from those elicited while viewing similar, but non-personal events.

fMRI studies have identified several brain regions that are often recruited when retrieving autobiographical memories. These include the medial and lateral prefrontal cortices (PFC), the lateral and medial temporal lobes (MTL, including the hippocampus and parahippocampal gyrus), the ventral parietal cortex, and posterior cingulate cortex [3,4,5]. These regions are typically co-activated to form separate neural networks, whose activation depends on the nature of the retrieved memory [6,7,8]. One of the primary neural networks that is recruited during autobiographical memory retrieval overlaps with the default-mode network, a set of neural structures that are co-active during passive resting state [9,10]. This includes two subcomponents – the medial prefrontal cortex (mPFC) network and the MTL network. The mPFC network includes the dorsal medial PFC, posterior cingulate, and ventral parietal cortices, and has been associated with self-referential processes during autobiographical memory retrieval [11,12,13]. The MTL network comprises the hippocampus, ventral medial PFC, retrosplenial, and ventral parietal cortices, and as been associated with recollection processes during memory retrieval [14,15]. A third network, the fronto-parietal network, includes the lateral PFC, anterior cingulate, and inferior parietal cortices and is thought to be associated with adaptive cognitive control processes [16,17], or an elaboration period of autobiographical memory retrieval that is implicated in the maintenance of episodic details in working memory [18].

The goal of the present study was to develop a novel paradigm that could be used to investigate rich, autobiographical memories without recourse to self-report. Accordingly, real-world, personal episodic experiences were captured in a naturalistic setting and played back to participants in an fMRI scanner a week later, along with similar and dissimilar experiences recorded by other participants. Participants were asked simply to view the videos in the scanner and make no overt behavioural response. A cross-validated, machine-learning model was trained on the fMRI data from all participants. Besides assessing the classifier’s accuracy levels, we also evaluated where in the brain our model drew most information for accurate classification. We hypothesized that the fronto-parietal network would be particularly important for differentiating between the video categories, given its documented involvement in the integration of internally directed thoughts and externally directed attention [19]. This network could play an important role in assessing, through self-projection, whether events belong to one’s own past, or that of another. In addition, we expected to see recruitment of the mPFC and MTL, given that these regions have been previously implicated in self-referential processes and autobiographical memory retrieval.

## 2. Materials and Methods

### 2.1. Participants

Twelve healthy volunteers participated in the experiment (six females, 20–34 years old, mean age = 25, SD = 4.3, all recruited from the Western University community). All healthy volunteers were right-handed, with normal or corrected-to-normal vision, and had no history of neurological disorders. They signed informed consent before participating and were remunerated for their time. Ethical approval was obtained from the Health Sciences Research Ethics Board and Psychology Research Ethics Board of Western University (IRB number: 00000940).

### 2.2. Procedure and Design

This study consisted of two phases: a *Mall-Visit Phase*, and an *fMRI Scan Phase*, carried out approximately one week apart.

### 2.3. Mall-Visit Phase

Participants were asked to sit in a wheelchair while a researcher pushed them around a local mall (Masonville Mall, London, ON, Canada). A small wearable camera (iON SnapCam LE, https://usa.ioncamera.com/snapcam/, see Figure 1A) was mounted on each participant, in order to capture autobiographical videos from the visit. Each participant visited two stores in the mall: the Apple Store, and The Bay. The recording session in each store lasted approximately 20 min. Participants were instructed to remain passive observers during the recording session—they were asked to refrain from moving their head or body while sitting in the wheelchair, and focus their attention on the events that happened in front of them (because the camera always pointed forward, looking sideways during a recording session would have meant potentially remembering events that were not captured by the camera). All participants visited the same two stores, but the specific route within each store was not pre-determined. Participants were also instructed not to talk or interact with other individuals during the recording session. The experimenter, however, was allowed to do so, where necessary (e.g., to ask members of the public to move aside).

### 2.4. fMRI Scan Phase

Participants were scanned approximately one week (mean = 6 days) following their mall visit recording session. Before each scanning session, the video recordings from each mall visit were segmented into 30-s clips. In the scanner, participants then viewed a randomized series of these clips that belonged to three different categories: *Own videos* (autobiographical videos, taken from their own experiences at the mall a week prior to scanning), *Other videos* (experiences of other participants in the study, recorded at the same locations), and *Bookstore videos* (recordings of other people’s experiences in a different location altogether; these recordings were collected in a similar manner to *Own* and *Other* videos, i.e., while sitting passively in a wheelchair), see Figure 1B. Participants were asked to simply view the videos in the scanner, and no overt behavioural response was required. Participants also wore earphones in the scanner as all of the videos included the audio that was recorded at the same time. Stimuli were presented using PsychoPy v1.85.4 [20].

There were six scanning runs in total (one participant completed only five runs due to fatigue), with each run comprising 16 videos, with a five-second break between them (Figure 1C). Each participant viewed 16 videos from each category, and each video was repeated twice during the scanning session, such that in total each participant, 16 × 3 × 2 = 96 videos. The first three runs comprised unique videos (16 *Own*, 16 *Other* and 16 *Bookstore* videos, presented in a random order), and the last three runs were the same videos presented earlier, repeated in a different order. Video clips that included any visible participant body parts or other obvious identifiers were removed from the stimulus set.

### 2.5. fMRI Data Acquisition

All neuroimaging data were acquired on a Siemens Tim Trio 3 Tesla MRI scanner at the Robarts Research Institute at Western University. A 32-channel head coil was used for all functional and anatomical scans. A high-resolution magnetization-prepared rapid-gradient echo (MP-RAGE) structural scan (repetition time (TR) = 2300 ms, 1 × 1 × 1 mm^3^, TE = 3 ms) was acquired at the beginning of each session and used for co-registration. Functional volumes were obtained with T2*- weighted whole-brain echo-planar imaging (EPI), sensitive to blood-oxygen-level dependent (BOLD) contrast. Each EPI volume consisted of 48 axial slices, acquired in an interleaved manner (TR = 1000 ms; TE = 30 ms; flip angle = 40°; FoV = 208 mm, voxel size = 2.5 × 2.5 × 2.5 mm^3^).

### 2.6. fMRI Data Preprocessing and Analysis

All fMRI preprocessing was conducted using the NiPype framework (version 0.13.1) [21], and SPM12 (http://www.fil.ion.ucl.ac.uk/spm/software/spm12/). The statistical analyses of the preprocessed data were performed in Python (version 2.7.13, www.python.org), using the libraries Nilearn; ‘0.3.1’, Numpy ‘1.13.1’, Nibabel ‘2.1.0’, and scikit-learn; ‘0.19.0’. In preparation for the machine-learning, the data were preprocessed, which included slice-timing correction, realignment of the data to the mean image, co-registration of functional and structural images, normalization to the Montreal Neurological Institute (MNI) template brain, and spatial smoothing with a 4 mm FWHM Gaussian Kernel. Before conducting the classification analysis, the BOLD signal from each video was averaged to yield a (number of videos × number of voxels) matrix for each participant.

### 2.7. Feature Selection and Classification

Two main analyses were conducted—*Leave one participant out cross-validation* (LOPOCV) and *within-participant cross-validation* (WPCV):

#### 2.7.1. Leave One Participant Out Cross-Validation (LOPOCV)

A cross-validated, Support Vector Machine (SVM) model with a linear kernel was employed for classification. Before each cross-validation, a sensitivity-based feature selection was performed; within the training data only, a voxel-by-voxel one-way ANOVA was performed to identify the voxels that show a differential response to the three conditions. The top 5% of voxels in the entire volume that had the strongest differential response to the categories (indicated by the largest F-value) were chosen as features for the training classifier. This approach is also referred to as “no peeking” ANOVA feature selection, as it is blind to whether the same voxels are differentially responsive in the testing set, avoiding a bias in the cross-validation. In addition, to ensure that any results obtained are significantly higher than those obtained from chance classification, the same analysis was repeated with randomly reshuffled category labels for each participant (for each participant, the mean of 10 analyses with randomly reshuffled labels was used). For all analyses, our main performance metric was classification accuracy. In addition, the results of this classification analysis are also reported in the form of a confusion matrix, reflecting the distribution of the classifier’s guesses for each of the video category types. Feature weights (i.e., importance maps) were derived using this analysis and were mapped back to a brain template for visualization purposes. With the *LOPOCV* approach, data from all participants were merged into a single dataframe, and a model was then trained 12 times (the number of participants), such that for each time, the training data comprised all but one participant (i.e., 11 participants) and was then tested on the data from the left-out participant. This process was repeated 12 times such that a single classification performance value was obtained for each participant.

#### 2.7.2. Within-Participant Cross-Validation (WPCV)

In order for the paradigm used in this study to be applicable for later testing of non-responsive, brain-injured patients, we also conducted a within-participant cross-validation analysis. With this approach, a separate machine-learning model is run for each participant. A five-fold cross-validation was used for assessing classification performance for each participant; that is, each time the classifier was trained on four folds (i.e., 80% of the data) and was then tested on the remaining one fold (20% of the data). This process was repeated five times, so that each fold was used once for testing. This allowed us to test the classification performance within a single participant—the main advantage here being that we did not assume that brain regions that are responsible for differentiating between the conditions are the same across participants.

## 3. Results

A series of analyses were performed to evaluate how accurately a whole-brain SVM classifier could discriminate between the fMRI activity associated with three naturalistic video categories. Importance maps were generated to determine the regions that were most diagnostic for these classifications.

### 3.1. Leave One Participant out Cross-Validation (LOPOCV)

The model was trained on 11 participants and was then tested on the remaining participant. This process was repeated 12 times, such that each participant was used as testing data once. Classification accuracies from all participants are presented in Figure 2A (blue dots). The mean accuracy across participants was 0.55 (SD = 0.09). In order to determine whether this was above chance, the same analysis was repeated with randomly reshuffled video labels. The mean classification accuracies from 10 permutations of this analysis with randomly reshuffled video labels are depicted in orange. The mean accuracy with the reshuffled labels across participants was 0.34 (SD = 0.02). A paired *t*-test between the two distributions (blue vs. orange dots) indicated that they were significantly different from each other (*t* = 8.35, *p* < 0.001). The results of this classification analysis are also reported in the form of a confusion matrix, reflecting the distribution of the classifier’s guesses for each of the video category types (Figure 2B). The mean feature importance maps from all participants revealed bilateral activity in the fronto-prietal network, dorsolateral prefrontal cortex, precuneus, mid-temporal regions, visual cortex, and fusiform gyrus (Figure 2C).

### 3.2. LOPOCV—Last Three Runs (Repeated Presentation of Each Video)

While viewing the videos for the first three runs (i.e., the first presentation of each video), participants might have actively tried to decide whether each video they viewed was a recording from their mall visit or not (despite not being told to do so). If that were the case, any observed differences might reflect the explicit (and successful or not) attempt to recall autobiographical memories, rather than the more passive “re-living” experience that more typically characterizes autobiographical recollection. Moreover, if the results relied upon effortful retrieval, they might be less applicable to behaviourally compromised individuals who might be capable of re-experiencing autobiographical memories, but unable to explicitly retrieve them. Hence, in this analysis we included only the last three runs, in which each video was presented for the second time and should therefore have been familiar to some extent. As all of the stimuli are now familiar, explicit attempts to decide which videos represented previously experienced events would be less likely to occur (as all videos had been experienced at this point), allowing participants to immerse themselves in the videos in a way that was more similar to ‘real world’ autobiographical memory. The mean classification accuracy using only the last 3 runs was 0.43 (SD = 0.10) (Appendix A). The same analysis with reshuffled video labels yielded a mean classification accuracy of 0.33 (SD = 0.02). A paired *t*-test between the two distributions indicated that they were significantly different from one another (*t* = 3.93, *p* < 0.005), indicating that the classifier was able to differentiate between the video categories even when all videos were somewhat familiar.

### 3.3. LOPOCV—Own vs. Other Videos Only

To rule out the possibility that the classification results obtained above were driven primarily by the *Bookstore* condition, which was entirely unfamiliar to all participants during the first viewing, we also ran an analysis with this condition removed. Thus, we were now left with the two most visually similar categories (videos captured at the same location), *Own* and *Other* conditions. Now, because there were only two conditions, chance performance was 50%. The mean classifier accuracy across participants was 0.56 (SD = 0.05). In order to determine whether this was above chance, the same analysis was repeated with randomly reshuffled video labels. The classification accuracies with the reshuffled labels are depicted in orange (Appendix A). The mean classifier accuracy across participants for these reshuffled labels was 0.49 (SD = 0.01). A paired *t*-test between the two distributions (blue dots vs. orange dots) indicated that they were significantly different from each other (*t* = 3.64, *p* < 0.005). See Appendix A for the other two contrasts (Own vs. Bookstore and Other vs. Bookstore).

### 3.4. An Analysis Restricted to Voxels from a Brain Mask Derived from a Meta-Analysis of Many Autobiographical Studies

To rule out the possibility that the results obtained above were driven primarily by visual cortex regions (i.e., visual differences between the videos), we also repeated the same analysis using a mask from Neurosynth, a platform for large-scale, meta-analysis of fMRI data from published studies (http://neurosynth.org/; [22]). A “reverse inference” map was generated, using the search term “autobiographical memory”, which yielded 84 studies that included 4225 activations. Based on the mask, the mean classification accuracy across participants was 0.43 (SD = 0.06). In order to determine whether this was above chance, the same analysis was repeated with randomly reshuffled video labels. The classification accuracies with the reshuffled labels are depicted in orange (Appendix A). The mean accuracy across participants was 0.34 (SD = 0.02). A paired *t*-test between the two distributions (blue dots vs. orange dots) indicated that they were significantly different from each other (*t* = 4.90, *p* < 0.001).

### 3.5. Within-Participant Cross-Validation (WPCV)

In this analysis, a separate model was trained and tested on the data of each participant in order to determine whether it was possible to classify autobiographical memories based on a single-participant’s data alone. A five-fold cross-validation was used for assessing the classification performance for each participant. Classification accuracies from all participants are presented in Figure 3A (blue dots). The mean accuracy across participants was 0.62 (SD = 0.12). In order to determine whether this was above chance, the same analysis was repeated with randomly reshuffled video labels. The mean classification accuracies from 10 permutations of this analysis with randomly reshuffled video labels are depicted in orange. The classification accuracies with the reshuffled labels are depicted in orange. The mean accuracy across participants was 0.30 (SD = 0.02). A paired *t*-test between the two distributions (blue vs. orange dots) indicated that they were significantly different from each other (*t* = 9.84, *p* < 0.001). The results of this classification analysis are also reported in the form of a confusion matrix, reflecting the distribution of the classifier’s guesses for each of the video category types (Figure 2B). Figure 3C shows brain regions that supported classification and were common across individuals. In this figure, importance voxels were aggregated, irrespective of their average positive or negative feature weight. Hence, this group importance map indicates the number of participants for which the voxel accurately distinguished between the different categories.

### 3.6. WPCV—Last Three Runs (Repeated Presentation of Each Video)

This analysis included solely the last three runs, in which each video was presented for the second time. The mean classification accuracy was 0.49 (SD = 0.09) (Appendix A). The same analysis with reshuffled video labels yielded a mean classification accuracy of 0.28 (SD = 0.03). A paired *t*-test between the two distributions indicated that they were significantly different from one another (*t* = 7.22, *p* < 0.001), demonstrating that within-participant video classification was possible when including only familiar video clips.

### 3.7. WPCV—Own vs. Other Videos Only

This analysis included only the two most visually similar categories—Own and Other. The mean accuracy across participants was 0.61 (SD = 0.14). In order to determine whether this was above chance, the same analysis was repeated with randomly reshuffled video labels. The classification accuracies with the reshuffled labels are depicted in orange (Appendix A). The mean accuracy across participants was 0.48 (SD = 0.04). A paired *t*-test between the two distributions (blue dots vs. orange dots) indicated that they were significantly different from each other (*t* = 2.99, *p* < 0.05), demonstrating that the WPCV classification results were not driven by the Bookstore condition, which included events that were the most visually dissimilar compared to participants’ autobiographical experiences from the mall visit.

### 3.8. An analysis Restricted to Voxels from a Brain Mask Derived from a Meta Analysis of Several Autobiographical Studies

This analysis was limited only to voxels included in the autobiographical memory brain mask acquired from the Neurosynth database. The mean accuracy across participants was 0.53 (SD = 0.12). In order to determine whether this was above chance, the same analysis was repeated with randomly reshuffled video labels. The classification accuracies with the reshuffled labels are depicted in orange (Appendix A). The mean accuracy across participants was 0.30 (SD = 0.02). A paired *t*-test between the two distributions (blue dots vs. orange dots) indicated that they were significantly different from each other (*t* = 7.15, *p* < 0.001).

## 4. Discussion

In this study, real-world autobiographical experiences were recorded using a wearable camera that was mounted on study participants while they were pushed in a wheelchair through a local mall. One week later, participants were scanned while they viewed a series of video clips; some were autobiographical and others were captured by other participants at the same, or a different location. Task demands were minimal; participants were asked simply to view the videos shown in the scanner and no overt behavioural response was required. A cross-validated, machine-learning model was trained on the fMRI data from all participants. The model was able to successfully classify the video categories above chance in all participants regardless of the classification method used, and the model feature-weights revealed that fronto-parietal regions, dorsolateral prefrontal cortex, precuneus, mid-temporal regions, visual cortex, and fusiform gyrus were particularly important for differentiating autobiographical experiences from non-personal experiences. Additional analyses revealed that classification accuracy was also above chance when (a) only video clips that were presented for a second time were included (thus, all three classes of video were familiar), (b) only the Own and Other conditions were used, meaning that the classifier could differentiate between true autobiographical memories and very similar footage involving the very same locations that was, nevertheless, not recorded during the participants’ trip to the mall and (c) a mask derived from a meta-analysis of several autobiographical memory studies was applied, demonstrating that video classification was successful even when adopting a hypothesis-driven approach derived from existing independent studies. This paradigm is able, therefore, to decode real-world autobiographical memories from fMRI data without requiring participants to respond overtly in any way.

In this study, the fronto-parietal network appeared to be particularly important for differentiating between autobiographical experiences and non-personal events. It has been suggested previously that this network may contribute to the integration of internally directed thoughts and externally directed attention [18,20]—that could be important when deciding, through self-projection, whether events belong to one’s own past, or that of another. In addition, Naci et al. [2] found that when participants attended to naturalistic stimuli evolving plot-driven movie clips, they displayed highly synchronized brain activity in supramodal frontal and parietal cortex and that this correlated with the degree of executive processing involved. The autobiographical movie clips used in this study might also be considered to involve a plot-driven element, although ‘the plot’ in this case involved events in one’s own life (e.g., remembering which route was taken, the individuals encountered on route, etc.). In addition, it has been suggested that this network is involved in the later retrieval stages of autobiographical memories (often referred to as the elaboration period), which involves the maintenance of episodic details in working memory, facilitating the sense of vividly re-experiencing a prior memory [19]. Thus, it is conceivable that this process occurred while participants were re-experiencing their autobiographical memories.

While the fronto-parietal network was clearly an important contributor to accurate classification in this study, the involvement of the mPFC and MTL networks was largely lacking. This was somewhat surprising, as mPFC activation has been linked with self-projection and mentalizing about the mind of others in previous fMRI studies (e.g., [23,24]). A potential explanation could be related to the nature of our task, which was not a traditional memory retrieval task, but rather, sought to emulate the experience of living and then reliving autobiographical memories as they occur in the real world. The majority of previous autobiographical memory studies have involved a protracted retrieval time that allows the integration of multiple cognitive processes that mediate recollection [4]. While our task allowed participants to immerse themselves in the videos in a way that was more similar to ‘real world’ autobiographical memory, it also made it difficult to separate the different stages of recollection, such as pre-retrieval (search), retrieval or recollection, and post-retrieval (monitoring and verification), all of which tend to recruit different networks (e.g., [13,18]). Thus, we conclude that the mPFC and MTL networks may have been recruited had our task required participants to overtly retrieve autobiographical experiences from their mall visit themselves (e.g., “imagine your mall visit from last week” or “decide if this video was taken while you visited the mall last week). While the approach used in previous studies may allow such deconstruction of the various processes involved in autobiographical memory, the current results suggest that these processes may not be central to how autobiographical memory occurs in the real world.

One might also wonder why we did not observe activity in the hippocampus, given its well-documented involvement in autobiographical memory (e.g., [25,26,27]). Other autobiographical memory studies typically require participants to make a memory judgment for each sequence of events depicted (strongly/moderately familiar/recalled [28]. Doing so, however, likely biases participants towards treating the paradigm as an explicit memory task, rather than allowing them to simply relive a previous experience. One possibility is that true autobiographical memory (in the sense that it occurs in the real world as a passive sense of familiarity when previously experienced events are ‘relived’), may not depend heavily on hippocampal involvement; rather, this important region may only be recruited when participants actively seek to review experienced events, or to make decisions about whether or not specific events have been previously experienced.

That being said, it is important to note that our findings were not driven solely by activity in regions of visual cortex either—it’s conceivable that if one or more conditions was too visually unusual then it might contribute disproportionately to the accuracy of the classifier. To ensure that this was not the case, we removed the visually dissimilar condition (Bookstore) from the analysis and still achieved above chance classification accuracy. Second, in an additional analysis, we used a mask derived from several autobiographical memory studies that did not include regions of the visual cortex, and accuracy was still above chance. We intentionally made the Own and Other conditions challenging, aiming for all the videos to be visually similar, but differing only in terms of the individual’s personal experience (e.g., the precise route taken in the store, people’s faces, and the voices that were heard). Changing some of these parameters—choosing to visit more stores in the mall, thereby increasing the visual variability between videos, or shortening the duration between the mall visit and the scanning session—would likely improve the classification accuracy of the model, although at a cost of making the classification less ecologically valid.

Because we wanted this experimental paradigm to be suitable for investigating autobiographical memory in patients who may be unable to physically navigate a real-world environment alone, or actively report on the contents of their memories, we made the task passive in two respects. First, during the recording session at the mall, all participants experienced events as passive observers—they were pushed in a wheelchair, and did not navigate the environment freely (i.e., they did not choose which sections of the store to explore, which routes to take, or whom to interact with). Some research suggests that active exploration may be more beneficial to memory. For example, in one study, active exploration (driving a virtual car) yielded enhanced recall of spatial information compared to passive exploration of the same environment (being a passenger in the same car [28]). It is therefore possible that allowing participants to navigate freely would have aided their memory and increased the classifier accuracy, or influenced the feature importance maps. Second, in the scanner, participants observed the videos passively, and no behavioural response was required (i.e., they were not asked to rate each video as being autobiographical or not). Pilot data from behavioural testing indicated that participants were indeed able to classify the autobiographical videos above chance. Thus, the decision to use a passive viewing task was made because we wanted them to fully immerse themselves in the videos and to not focus on solving a cognitive challenge that might have confounded the autobiographical experience (e.g., pressing a button in the middle of a video to indicate that they recognize it as being autobiographical). Asking participants to classify videos in the scanner would likely alter the nature of the task, perhaps causing participants to shift their attention once they “solve” the challenge.

The mean classification accuracies for Leave One Participant Out Cross-Validation (LOPOCV) and Within-Participant Cross-Validation (WPCV) were 0.55 and 0.62, respective. While these three-way classification accuracies are not high in and of themselves, they are significantly higher than what would be expected by chance (see also [29]). It is also worth noting that the classifier’s performance will depend of several factors, such as the complexity of the task (e.g., it is easier to achieve higher accuracy in a simple motor task compared to a high level memory task [30], the ML algorithm used—here we used SVM, but other algorithms may achieve a higher classification accuracy (but this can come at a cost in terms of the explainability of the results).

In this investigation we utilized an experimental paradigm that is suitable for investigating autobiographical memory in patients who may be unable to physically navigate a real-world environment, make overt responses and/or follow complex task instructions. While this was our goal, it can also potentially be a limitation—for example, if one’s goal is to maximize fMRI classification accuracy. Making the task active rather than passive is one factor that could help in this regard—participants could navigate the wheelchair themselves, or if testing patients in not planned, researchers can forgo using a wheelchair altogether. Finally, researchers can collect behavioural data, in addition to neuroimaging data—in the scanner, participants can indicate after the presentation of each video clip, whether they recall that event or not. Another suggestion is to control for the number of unique faces/structures that appear in each video category, in order to better understand whether this factor plays a role in video memorability, and representation in the brain (classifier feature weights).

## 5. Conclusions

In this study, our goal was to investigate autobiographical memory by attempting to decode when participants were re-experiencing an entire event, captured on video from a first-person perspective, relative to a very similar event experienced by someone else. Although each participant had unique autobiographical memory experiences during the mall visit, a single classifier model trained on the data from all participants was able to successfully classify autobiographical and non-autobiographical videos, suggesting that there is a shared mechanism for processing/storing these memories. The paradigm introduced here allows researchers to investigate personal memories without requiring any overt response or any complex task instructions. This makes it ideal for investigating autobiographical memory in patients with brain injuries who may be unable to overtly respond to experimental stimuli.

## Figures and Tables

**Figure 1 brainsci-11-00521-f001:**
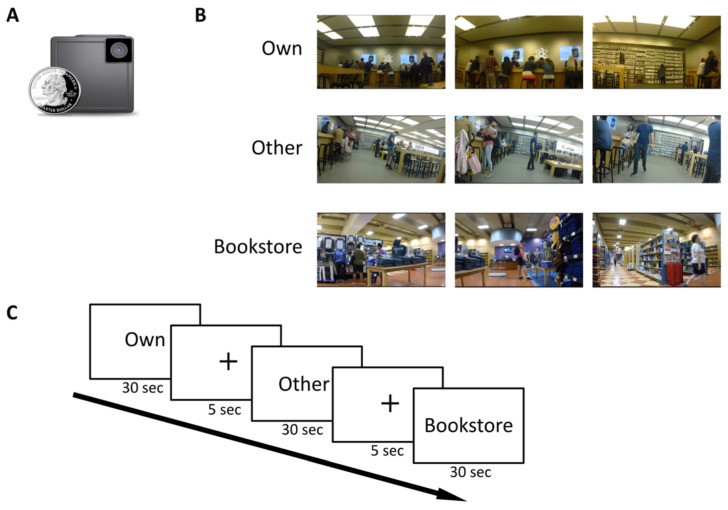
Experimental design. (**A**) The small wearable camera used in the study (iON SnapCam LE). (**B**) Participants viewed short video clips (30 sec long) in the scanner, that belonged to one of three different categories: Own videos (autobiographical videos, taken from their own experiences at the mall a week prior to scanning), Other videos (experiences of other participants in the study, recorded at the same location), and Bookstore videos (recordings of other people’s experiences in a different location altogether). (**C**) Video clips were presented in a random order, with a five second ITI. Each scanning run comprised 16 videos. There were six runs in total. Each participant viewed 96 clips in total (48 clips each repeated once).

**Figure 2 brainsci-11-00521-f002:**
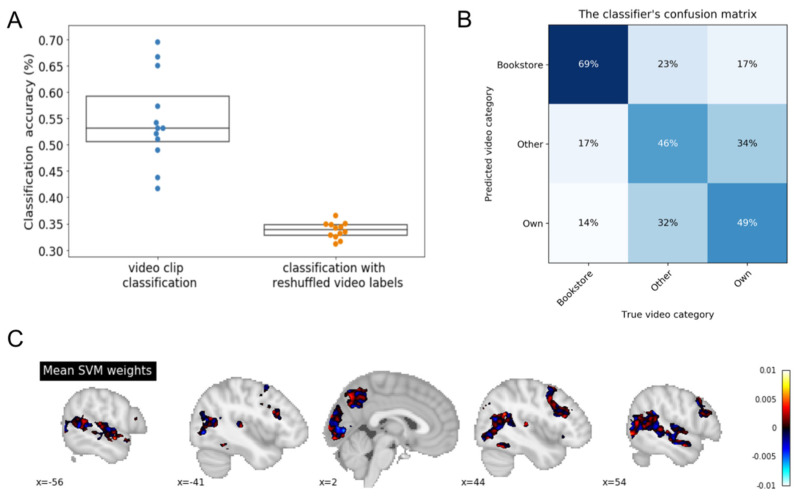
Leave-one-participant-out CV analysis. (**A**) Classification accuracies for the 12 participants (in blue). The model was trained on 11 participants and was tested on a single participant. This process was repeated 12 times such that each participant’s data served as test dataset once. As a control, the same procedure was repeated 10 times with randomly-reshuffled video labels (mean chance-classification accuracy for each participant in orange). (**B**) Confusion matrix for the three-way classifications. Percentages indicate the distribution of the classifier’s guesses for each video category. Shown here is the average confusion matrix across the 12 participants. (**C**) Average feature importance maps of the classifier (all data). Fronto-prietal regions, dorsolateral prefrontal cortex, precuneus, mid-temporal regions, visual cortex, and fusiform gyrus were particularly diagnostic in differentiating autobiographical experiences and non-personal experiences.

**Figure 3 brainsci-11-00521-f003:**
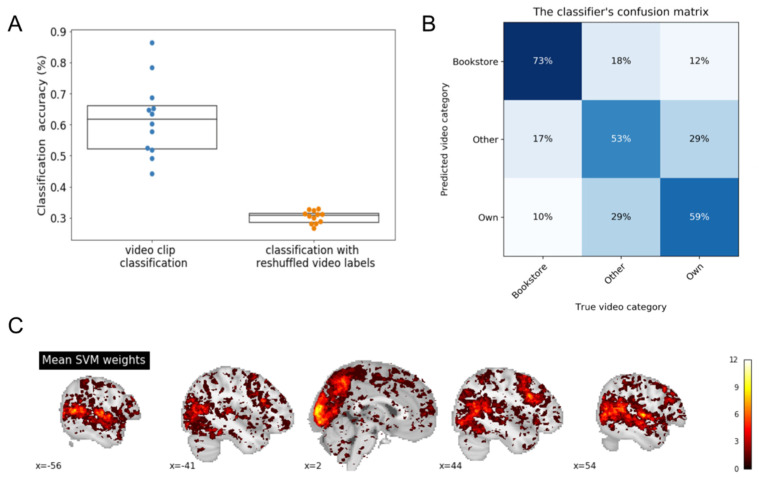
Within-Participant Cross-Validation (WPCV) analysis. (**A**) Classification accuracies for the 12 participants (in blue). 12 models were trained and tested on the data of each participant separately, using a five-fold CV. As a control, the same procedure was repeated 10 times with randomly-reshuffled video labels labels (mean chance-classification accuracy for each participant in orange). (**B**) Confusion matrix for the three-way classifications. Percentages indicate the distribution of the classifier’s guesses for each video category. Shown here is the average confusion matrix across the 12 participants. (**C**) Brain regions that supported classification across individuals. Importance voxels were aggregated, irrespective of their average positive or negative feature weight. Hence, this group importance map indicates the number of participants for which each voxel accurately distinguished between the different categories. Colorbar indicates the number of participants. Regions that were diagnostic in differentiating between categories were similar to the LOPOCV analysis and included fronto-prietal network, dorsolateral prefrontal cortex, precuneus, mid-temporal cortex, visual cortex, and fusiform gyrus.

## Data Availability

Data available on request due to privacy and ethical restrictions.

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
