# Peer review of "Differentiating Real-World Autobiographical Experiences without Recourse to Behaviour"

_brainsci, 2021, doi:10.3390/brainsci11040521_

Round 1
Reviewer 1 Report
This manuscript contains some very novel results concerning fMRI of autobiographical data. Here are a few suggestions to help improve the manuscript.
1) In the methods section, add the number of fMRI TR periods for each type of fMRI scan.
2) Add a figure which shows a scatter plot comparing the fMRI activation zscore from an important brain region for autobiographical period versus non-autobiographical period.
3) Add a paragraph in the discussion comparing your % accuracy for classification versus other fMRI classification from previous publications. Does this manuscript report % accuracy above or below other fMRI classification reports?
4)The authors talk about applying these methods to patients with brain injury. How would you design the scan/task to best test an patient with brain injury?
Reviewer 2 Report
Erez, Gagnon and Owenpresent a nice study where they attempt to decode when participants were re-experiencing an entire event, captured on video from a first-person perspective, relative to a very similar event experienced by someone else. A machine-learning model was able to successfully classify the video categories above chance, both within and across participants. The authors find a set of FrontoParietal brain regions that codes autobiographical experiences from non-autobiographical ones with high accuracy.
I congratulate the authors for this nice experiment and the soundness of the methods very informative.
The only recommendation I would make is to include a limitations section and include future ideas for improving this type of experiments.
